# Evaluation of a Community-Led Intervention in South London: How Much Standardization Is Possible?

**DOI:** 10.3390/ijerph17072523

**Published:** 2020-04-07

**Authors:** Derek Bolton, Nina Khazaezadeh, Ewan Carr, Matthew Bolton, Eirini Platsa, Imogen Moore-Shelley, Ana Luderowski, Jill Demilew, June Brown

**Affiliations:** 1Department of Psychology, Institute of Psychiatry, Psychology and Neuroscience, King’s College London, London SE5 8AF, UK; analuderowski@gmail.com (A.L.); June.Brown@kcl.ac.uk (J.B.); 2Maternity Services, Guy’s Hospital, Guys and St Thomas’s NHS Foundation Trust, Great Maze Pond, London SE1 9RT, UK; Nina.Khazaezadeh@gstt.nhs.uk (N.K.); eirini.platsa@gstt.nhs.uk (E.P.); 3Department of Biostatistics and Health Informatics, Institute of Psychiatry, Psychology and Neuroscience, King’s College London, London SE5 8AF, UK; ewan.carr@kcl.ac.uk; 4Citizens UK, 112 Cavell Street, London E1 2JA, UK; Matthew.Bolton@citizensuk.org (M.B.); imogen@connectionsinmind.co.uk (I.M.-S.); 5Maternity Services, King’s College Hospital, King’s College Hospital NHS Foundation Trust, Denmark Hill, London SE5 9RS, UK; jill.demilew@btinternet.com

**Keywords:** community health, health inequalities, community engagement, community organizing, complex interventions, hierarchy of evidence, PACT, Citizens UK, evaluation, methodology

## Abstract

It is widely recognized that public health interventions benefit from community engagement and leadership, yet there are challenges to evaluating complex, community-led interventions assuming hierarchies of evidence derived from laboratory experimentation and clinical trials. Particular challenges include, first, the inconsistency of the intervention across sites and, second, the absence of researcher control over the sampling frame and methodology. This report highlights these challenges as they played out in the evaluation of a community-organized health project in South London. The project aimed to benefit maternal mental health, health literacy, and social capital, and especially to engage local populations known to have reduced contact with statutory services. We evaluated the project using two studies with different designs, sampling frames, and methodologies. In one, the sampling frame and methodology were under community control, permitting a comparison of change in outcomes before and after participation in the project. In the other, the sampling frame and methodology were under researcher control, permitting a case-control design. The two evaluations led to different results, however: participants in the community-controlled study showed benefits, while participants in the researcher-controlled study did not. The principal conclusions are that while there are severe challenges to evaluating a community-led health intervention using a controlled design, the measurement of pre-/post-participation changes in well-defined health outcomes should typically be a minimum evaluation requirement, and confidence in attributing causation of any positive changes to participation can be increased by use of interventions in the project and in the engagement process itself that have a credible theoretical and empirical basis.

## 1. Introduction

The importance of involving communities in their own health and the importance of co-production and community control for promoting engagement are widely recognized [1,2,3]. ‘Community-controlled’ or ‘community-led’ are distinct from ‘community-based’, which refers to interventions led by health professionals but in community settings. The importance of evaluating novel community health interventions is also widely recognized, to influence adoption decisions by health providers, funders, and policymakers [4,5]. However, there are challenges to evaluating so-called ’complex interventions’, which include community health interventions, using hierarchies of evidence derived from laboratory experimentation or clinical trials. ‘Complex interventions’ can be defined in various ways [6,7,8]. The simplest is in terms of the number of components (e.g., a course of psychotherapy is complex compared with a course of pharmacotherapy). Another sense arises from the system into which the intervention is introduced and the extent to which associated contextual parameters moderate the effect of the intervention. Community health projects are “complex” in both senses: they are typically combinations of interventions, and they are implemented in complex and variable local settings [6,7,8,9].

Regarding evaluation strategies, the concept of ‘hierarchy of evidence’ is prominent in health literature, ranking evaluation designs according to the strength of evidence they are capable of providing [10,11,12,13]. The guiding principle is that, as one ascends the hierarchy, we increase the degree of confidence that changes are caused by the intervention, so-called ‘internal validity’. The hierarchy typically ranks randomized controlled designs as providing the strongest type of evidence, followed by non-randomized controlled designs (sometimes called ‘quasi-experimental’ designs), followed by various uncontrolled designs, including simple pre-/post-intervention assessments of the outcome of interest in a single group receiving the intervention [10,11,12,13].

There is an active debate in the literature as to whether hierarchies of evidence derived from laboratory experimentation and clinical trials are optimal for evaluating complex interventions in community health [7,8,9,14,15]. The main challenges revolve around the requirements for evaluation designs higher in the hierarchy to ensure consistency, sometimes called ‘standardization’, in turn requiring researcher control. These requirements increase confidence that the intervention, and its effects on those receiving it, are well-defined, generalizable, and replicable. There are therefore two main aspects of consistency, standardization, and researcher control: (i) the consistency of the intervention across applications, and (ii) control over the sampling frame and sampling methodology, enabling clear sample definition, generalizability, and the construction of control and intervention groups.

In their paper ‘Complex interventions: how “out of control” can a randomized controlled trial be?’, Hawe and colleagues [16] focus on the problem of ensuring the consistency of a community health intervention across applications. They note that the application and effects of complex community health interventions may be affected by variable local factors, and that seeking to standardize the intervention across applications may be sub-optimal compared with seeking to assess relevant factors in different localities and adapting the intervention accordingly. Hawe and colleagues [16] suggest a novel solution, in which there is a both standardization of a general model for the intervention, and adaptation to local conditions, hence reconciling the standardization requirement for randomization designs with the complexity of public health interventions.

This approach can be seen in psychological therapy. Clinical researchers develop a general model of the causal factors at work in the condition of interest, identifying targets for intervention, together with a description of procedural intervention paradigms to address these targets. While the model as a whole is hypothesized to apply across all individuals who have the condition, it is acknowledged that there are relevant individual differences that have to be assessed and incorporated into the implementation of the general model. This approach has been used for example in cognitive therapy models and treatment trials of panic disorder [17], post-traumatic stress disorder, [18], and obsessive compulsive disorder [19].

The second main aspect of consistency and standardization, and the central topic of this paper, is control over the sampling frame and sampling methodology, enabling clear sample definition and the construction of control and intervention groups, either by randomization or using a case-control design. Community health projects present several challenges to this kind of standardization. One challenge can occur when the total group in the evaluation is defined but the researcher is not in control of assignment to condition. For example, a research participant initially assigned to the control group chooses to join in the intervention group. Another more fundamental challenge arises when the researcher is not even in control of defining the sampling frame, sampling method, or resulting group, as is often the case in community-organized and community-led projects. Here, not only is the assignment of participants to conditions out of researcher control, but the construction of the participant group itself is also out of control. This means that fundamental evaluation concepts, such as *sampling*, *population*, and *matched control*, lose meaning. To construct a controlled evaluation of a community-led health intervention requires putting the researcher back in control of population definition and sampling strategy, contradicting the community-led approach. Thus, researcher control over sampling strategy is incompatible with community-led studies. As noted above, the challenge of the standardization of the intervention to enable robust evaluation is recognized in the literature, but this further challenge of standardizing the sampling strategy in community-led projects has received less attention. The problem is important because it signals a tension between two highly-valued and established paradigms: community leadership and control to maximize engagement in community health projects to, for example, address local health inequalities, and researcher leadership and control to maximize the strength of evidence by controlled evaluation. This tension is the core problem addressed in this paper.

We consider these issues as they played out in a community-organized and led health project in South London—the PACT project, Parents and Communities Together. PACT involved collaboration and co-production between communities, health professionals, and clinical academic researchers. The key components of the project were social support and health education aimed at improving maternal mental health and health literacy and thereby child health and developmental outcomes. PACT was built on a successful pilot study assessing the feasibility of community organizing to engage local mothers [20]. The pilot study used a repeated measures design with assessments pre- and post-participation in the project. For the larger main study, we had intended to carry out the evaluation with a controlled design to achieve a higher grade of evidence, important in its own right but also, we supposed, increasing the chances of adoption by health service commissioners. This intention was reinforced by the fact that the funding application form included many sections on increasing the likelihood of future adoption and one section headed “Describe how the evaluation will demonstrate that the anticipated benefits are attributable to the work of the project rather than other factors”. We had also been long aware, however, that there were substantial challenges in constructing a control group for a participant group engaged by community-led methods, and, to work around this problem, we were led to planning two simultaneous evaluation studies: a case-control study, for mothers engaged by research procedures, but also a simpler, uncontrolled, repeated measures (pre-/post) design, for mothers engaged by community-led methods. The decision-making processes are described below.

Even before planning the pilot study, we were aware that community organized engagement methods, because of their flexibility, would make the construction of a control group highly problematic. To illustrate, the community engagement methods that emerged in PACT included membership of participating institutions, such as faith groups and schools, some involved from the start, some joining along the way, engagement by volunteer “parent champions”, invitations to mothers attending a local baby clothes bank, and being brought along by friends and neighbors. In short, such community-led engagement tends to be open-ended, improvised, opportunistic, with accordingly unpredictable results in terms of the characteristics of those who come to participate. We recognized that the flexibility of community-organized engagement and the resulting unpredictability of the eventual sample made constructing a control group problematic, and we approached the evaluation strategy section of the application for funding for the PACT project with these issues in mind.

First, we considered the design at the top of the evidence hierarchy, randomization. We noted that one theoretically plausible way to construct control and intervention groups would be individual randomization following community engagement. However, we noted, this research strategy would come into conflict with the aim of the project to maximize engagement. In more detail, the aim of the project was to provide a new service for local mothers, and community-led engagement was employed in order to maximize access and participation. Community leaders were entirely unwilling—as were the researchers—to envisage the idea of gate-keeping that would be required by an individual randomized design. Mothers who had been encouraged to join a fun and helpful community project would, depending on allocation to treatment vs. the control condition, be turned away at the gate. Individual randomization would have required changing the engagement procedure to remove the community-led engagement, contrary to the project aim of maximizing and widening participation.

Another approach to evaluation would be cluster randomization. Cluster randomization would partly solve the problem insofar as prospective participants in control clusters would never be promised inclusion in the intervention. However, by the same token, participants in control clusters would not be engaged by community-organized and community-led processes, leading to different recruitment methods in intervention and control clusters. This further complication could be avoided with an active control group (e.g., another activity distinct from the intervention where community-led engagement would also be appropriate) but the nature of such an intervention is unclear and this design would require higher statistical power to detect differences. This links with another consideration, which is cost. Cluster randomization requires many sites and are accordingly very expensive. They are more suited for proposal to a national research funder than the local innovative service funder we were applying to. We therefore concluded that cluster randomization for evaluating PACT was unsuitable for the principled and pragmatic reasons listed above.

Having reasoned that individual and cluster randomization designs were not feasible, we opted for a case-control design based on two geographically separated but demographically comparable areas in the same borough. Given the aim of benefitting maternal mental health and thereby child outcomes, women in pregnancy was a sensible sampling frame, with recruitment in maternity health clinics. We were aware, however, that by going down this route we were solving some problems but creating others. We could maintain the inclusivity of the PACT project by accepting all-comers engaged by community-led methods, but, for the controlled design, we accepted that we had to establish what was in effect a separate sub-project comprising participants who, while taking part in the same intervention, were recruited into a research project by processes under researcher control. Hence, we planned two simultaneous evaluation studies: a case-control study, for mothers engaged by research procedures, but also a simpler, uncontrolled, repeated measures (pre-/post) design, for mothers engaged by community-led procedures.

The principal finding that we focus on in this paper is that these two evaluation studies had completely different results: the community-engaged sample showed improvements over time, while the participants in the intervention arm of the researcher-engaged sample did not. This finding is actually not connected with having a control group. Rather, it concerns two groups, both of which participated in the intervention, one of which showed improvements and the other did not. We address three questions: First, what explains the different patterns of results? Second, which was the appropriate evaluation strategy to test the effectiveness of the PACT project? Third, are there general implications for evaluating community-led health projects? The principal conclusions are that while there are severe challenges to evaluating a community-led health intervention using a controlled design, the measurement of pre-/post-participation changes in well-defined health outcomes should typically be a minimum evaluation requirement, and confidence in attributing causation of any positive changes to participation can be increased by using interventions in the project and in the engagement process itself that have a credible theoretical and empirical basis.

Section 2 presents PACT as a case study illustrating the challenges of using controlled designs for evaluating the effectiveness of community-led health interventions. Section 3 describes the differences between the results of the controlled and uncontrolled evaluations. Further details of the two evaluation studies are provided elsewhere: the case-control study and its negative findings in Appendix A; and details for the simpler evaluation with a community-engaged sample and its positive findings are in a separate report [21]. We used mixed methods to evaluate PACT, recognizing the value of qualitive as well as quantitative approaches in health research [22]; we focus on quantitative results here and report the findings of the qualitive study elsewhere [23].

## 2. The PACT Case Study: A Community-Led Health Project Evaluated Using Two Strategies

### 2.1. PACT: Aims, Objectives, Collaborations, and Setting

The Parents and Communities Together (PACT) project was implemented in the South London borough of Southwark between 2016–18. PACT aimed to apply the findings of developmental health science, and associated prevention technologies, specifically social support and health education, to improve the health outcomes of new mothers and their children. The health science studies and the associated policy context underpinning these aims are substantial, e.g., [1,3,4,24,25,26,27], but it is not our objective to review them here. The main objectives were to make opportunities for local communities to become involved in their own health, specifically maternal and child health, to maximize engagement through community organizing and leadership, to address local health inequalities, to increase community capacity and social capital, and to enhance relationships between communities and local statutory health services and communities. There was substantial preparatory work from around 2010, including a pilot project in 2013–14 [20] and a co-production phase in 2014–15. Since completion in 2018, the methodology and interventions have been adopted and rolled out to other sites in London and across the UK.

The PACT project involved collaboration between Citizens UK [28], the largest community organizing charity in the UK, Citizens UK local member institutions, and King’s Health Partners, an Academic Health Sciences Centre [29]. Citizens UK uses the ‘broad-based’ community organizing model and methodology, which is well theorized, deriving from the work of Saul Alinsky in Chicago [30,31] and which is applied by community organizations throughout the U.S. and by Citizens UK in the UK. The key features of the approach include building trust-based reciprocal relationships among individuals in already existing communities, particularly civic institutions, fostering networks among diverse institutions, developing community leadership, and working towards goals decided by communities. Citizens UK employs paid, trained professional community organizers who work with volunteer community leaders.

PACT was set in an inner-city London borough, focusing on two electoral wards with high levels of social deprivation and immigration. The intervention was based in three local hubs: one church, one church-related center, and one community center. It was organized by four paid part-time staff: a community organizer/project manager, a health visitor, and two group leaders who were local mothers. In addition, there were five volunteers at a hub at any one session to help with childcare. In addition, there were local parent champions who were parents and/or key individuals from civic organizations (e.g., a children’s center, local primary school, churches, and mosques) and community organizations (e.g., charities) who were trained to take on a new role in their local community, speaking to other parents and signposting them to local services in the borough, including the PACT project.

### 2.2. PACT: Community Engagement, “Interventions”, Hypotheses, and Measures

PACT community engagement methods used the “broad-based” community organizing model, as noted above, involving relationships in and among civic institutions. Institutions involved in the PACT project, from the beginning or joining later, included churches, mosques, schools, health visiting and midwifery clinics, and Children’s Centers. Mothers were invited to join in the project or self-referred after signposting by members of these institutions, often by the parent champions, or by mothers already engaged in the project, promoting the benefits of using such services. Leaflets were posted and distributed in participating organizations publicizing the project and encouraging attendance. Mothers engaged in the project encouraged friends and neighbors to join in and made a special effort to meet isolated women that they knew of and encouraged them to attend. Similarly, health visitors and midwives especially encouraged women who appeared isolated to attend the project and arranged for them to be welcomed. One of the main meeting places also had a “Baby Bank”, where child clothing, equipment, and accessories were donated and given to mothers in need; mothers visiting this facility were welcomed by mothers already involved with PACT and were encouraged to join in.

The core PACT interventions were social support and health education. Social support was provided by the mothers themselves in regular weekly meetings. Health education workshops were co-designed and co-delivered by the mothers working with the health visitor on the project, bringing in other health professionals (such as midwives, child psychologists, dietitians, and parent trainers) as needed. Further details of the interventions are in Appendix A. Both of these interventions emerged in the pilot project and grew in the co-production phase preparing for the main PACT project. One of the first decisions in the co-production phase taken by the participating mothers was to rename the project the Parents and Communities Together (PACT) project, rejecting the previous name, suggested by the clinical academics for the pilot project (Strengthening Babies’ Futures). Also of note in the co-production phase was an on-going muddle about the term “intervention” (which is why we have it in quotes in this sub-heading). The health professionals could not think without it, while lay participants did not see much sense in it. This was probably more than a terminological issue, more a marker of distinctive cultures and concepts of the health sector on the one hand and lay communities on the other. An “intervention” is another name in the health sector for, roughly, treatment, something provided to patients by health services, while what the community stakeholders had in mind was people doing for themselves together, with or without the help of health professionals. The general point about distinctive cultures and concepts will be picked up in the Discussion in the context of considering background assumptions of the applicability of clinical trial methodology.

We hypothesized that social support and health education would improve maternal mental health and health literacy, and thereby, for new mothers, infant health and developmental outcomes in the first 12 months. We assumed that the interventions would target specific relevant causal pathways: first, social support, a form of social capital, reduces maternal distress and improves parenting self-efficacy and reflective functioning, thereby benefiting infant development and early child health outcomes; second, that health education of mothers in pregnancy and infancy increases health literacy, thereby benefiting infant health and developmental outcomes. The science underpinning these models of causal pathways and the interventions targeting them is well reviewed in the policy documents already cited [1,3,22,23,24,25], and is not reiterated here. We were translating the models into practice, not testing them, and the evaluation was not intended or set up to test them.

### 2.3. Evaluation: Using Two Strategies

#### 2.3.1. Rationale for Two Strategies

For the reasons reviewed in the Introduction, we used two strategies to evaluate PACT: first, a quasi-experimental case-control design that would increase the strength of evidence obtained, but which required replacing community-led engagement with a researcher-controlled sampling pool and sampling; second, a design that retained community-led engagement but at the expense of being controlled. The use of a case-control design to evaluate PACT was confirmed as a sensible choice in consultation with the National Institute of Heath Research (NIHR) Design Service, a service offered by the research division of the NHS [32].

#### 2.3.2. Evaluation Study #1—The “Case-Control Study”

The sampling frame was pregnant women at around 22 weeks gestation consecutively attending NHS community maternity clinics in two geographically separated patches in Southwark (Camberwell for the intervention arm and Bermondsey for the control arm). Recruitment was carried out by research midwives, following strict NHS research ethics practice. In clinics in the intervention area, potential participants were invited to take part in the PACT project and evaluation by completing questionnaires and interviews. In clinics in the control area, participants were invited to take part in the evaluation only. No non-NHS staff were permitted contact with the patients until after informed consent was given; community organizers, leaders, and other community participants in the PACT project had no place in the recruitment procedures. The inclusion criteria for the case-control study were: being over 18 and the ability to speak sufficient English to complete the questionnaires. In addition, there was an exclusion criterion dictated by current NHS guidance, mainly suitability for referral to specialist services (e.g., being below 19 years of age, currently suffering with depression, or currently in treatment in perinatal mental health services), and we also excluded mothers living in postcodes that were sampling pools for a similar project. On the basis of power calculations, we aimed to include 67 mothers in each arm of the evaluation, totaling 134 (for details, see Appendix A). All mothers eligible for inclusion in the evaluation attending maternity clinics in each patch consecutively were invited to take part in the evaluation study, and recruitment closed when the target numbers were reached. All participants taking part in the evaluation were offered compensation in the form of £30 vouchers on each occasion they completed assessments. The study was reviewed and favourable opinion given by the NHS National Research Ethics Service Committee London–Fulham, REC reference 15/LO/1227, ID 182843. The study was registered in the ISRCTN Register at http://www.isrctn.com/ISRCTN21987651.

In both evaluations, self-reported demographic details were recorded, including age, place of birth, ethnicity, first language, occupation, relationship status, and the number of children. In both studies, the primary health outcome was maternal mental health, assessed in both studies using two measures:

The Patient Health Questionnaire (PHQ-9) is a widely used and reliable 9-item measure, which assesses Major Depressive Disorder [33]. A 3-point Likert scale is used: not at all (0), several days (1), more than half the days (2), and almost all the time (3). Scores of 5, 10, and 15 are the cut-offs for mild, moderate, and severe levels of depression [33].

The Generalized Anxiety Disorder Questionnaire (GAD-7) is a widely used and reliable 7-item measure that assesses Generalized Anxiety Disorder [34]. A 3-point Likert scale is used: not at all (0), several days (1), more than half the days (2), and almost all the time (3). Scores of 5, 10, and 15 are the cut-offs for mild, moderate, and severe levels of anxiety [34].

Other outcomes included maternal health literacy and social capital/social support, and infant health and developmental outcomes at 12 months (see Appendix A for details). Maternal assessments were made at recruitment into the evaluation, around 22 weeks into pregnancy, and at follow-up 6 months post-natal, approximately 10.5 months after baseline. Infant physical development was assessed through the first 12 months, and infant social and emotional development was assessed at 12 months.

#### 2.3.3. Evaluation Study #2—The “Community Evaluation Study”

The sampling frame for the second evaluation study was local mothers participating in PACT engaged by community organizing methodology, as described in Section 2.2. During the three years of the PACT project described here, 425 local mothers participated (including participants in the case-control study intervention arm). On the basis of power calculations and other considerations—detailed in Appendix A—we aimed to include approximately 60 mothers in the evaluation. Participants in the evaluation were selected from the 425 participants in PACT using the following inclusion criteria: being over 18, female, the parent of at least 1 child, the ability to speak sufficient English to complete the questionnaires, and being engaged in the PACT project (by community-engagement routes) but for less than 2 months. These inclusion criteria, to emphasize, refer to selection for the evaluation, not to participation in the PACT project itself. All mothers eligible for inclusion in the evaluation engaging consecutively in the PACT project from its beginning were invited to take part in the evaluation study and recruitment closed when the target number was reached (at 61). Contact was made by named individuals, in practice research assistants who were also engaged in the project. The same self-reported demographic details were recorded and the same maternal mental health measures were used as for the case-control study detailed above. Other main outcomes included maternal health literacy and social capital/social support, and details of the measures used are given in Appendix A. As a compensation for the participants’ time at baseline and follow-up assessments, £30 shopping vouchers were offered. Assessments were made at baseline, following informed consent to take part in the evaluation, and at follow-up six months later. Ethical approval was given by the King’s College London Research Ethics Committee, REC Reference number HR15/162334.

## 3. Results

### 3.1. Main Findings: Unexpected Differences from the Two Evaluation Studies on the Primary Health Outcome

The results of the two evaluation studies were broadly negative for the case-control study, but broadly positive for the community study. We focus here on differences in the primary outcome, maternal mental health, assessed by the PHQ-9 and GAD-7. Table 1 shows for both studies descriptive statistics for these measures at baseline and follow-up with a paired t-test significance level of differences.

The main points of interest in the pre-post data shown in Table 1 are as follows:The community study sample showed predicted significant improvement on both measures;The case-control study intervention arm sample showed no significant change on either measure; predicted improvement was not found;The case-control study control group showed no change on the GAD-7, consistent with prediction, but showed an unexpected significant improvement on the PHQ-9;There were large differences between the baseline scores of the two samples in the two studies on these primary health outcome measures, indicating that with respect to the primary health outcome variable, the two studies had different samples. The community sample had a higher baseline mean on both measures. In unplanned analyses, we examined the extent of these baseline differences between the community study sample and the case-control study total sample, finding that differences were significant for both measures PHQ-9 (*p* = 0.001; t = 3.36, df = 57) and GAD-7 (*p* = 0.001; t = 3.36, df = 57).

Table 1 shows completer analyses in the community study and intention-to-treat (ITT) analyses in the case-control study. These were planned primary analyses. Secondary planned analyses, using ITT analyses in both, and completer analyses in both, produced consistent patterns of results.

We also continued with planned more sophisticated analyses of the case-control mental health data permitted by the design: regression analyses to examine within and between group changes through time, taking into account confounders, such as baseline scores and the levels of deprivation. However, we were clear that the data in Table 1 were evidently unpromising and that more sophisticated analyses were probably not going to reveal any support for the hypotheses under test. This was confirmed when we did the further analyses, as reported in Appendix A.

### 3.2. Other Findings

The findings on other measures in the two evaluation studies were broadly in line with the results for maternal mental health. In the community evaluation study, there was evidence of improvement (see results reported in full elsewhere [21]). In the case-control study, the intervention group showed no improvements on other measures, usually none within-group, or occasionally, and in the context of multiple testing, there were within-group changes in some subscales, but no more than in the control group. The case-control study also assessed infant outcomes at 12 months, which showed broadly no differences between intervention and control group infants. There were a few statistically significant differences in scales or subscales, in the predicted direction, but we have no confidence in them for several reasons: first, they emerged in the context of multiple comparisons, raising the likelihood of chance findings; second, and consistent with that, some apparently conflicted with negative findings on other scales or subscales, and, thirdly, in the absence of any detected difference in maternal mental health, health literacy or social capital we had no support for the hypothesized mechanisms leading to an improvement in infant outcomes.

## 4. Discussion

### 4.1. Main Finding: The Two Evaluation Strategies Led to Differing Patterns of Results

Two strategies were used to evaluate the benefits of participation in the PACT community health project, involving distinct sampling pools and engagement methods. The two studies showed different patterns of pre-post results: the sample of community-engaged participants in PACT showed significant improvement in mental health assessments between baseline and follow-up, whereas the research-engaged intervention group in the case-control study did not.

We should emphasize that the problem in this pattern of findings is nothing to do with the behaviour of the control group in the case-control study. The important point is not the controlled effect size in the case-control study, but is rather the uncontrolled effect size; the finding was not that *both* the intervention *and* control groups saw improvement. Nor is the problem that the control group improved significantly on one of the mental health measures (PHQ-9; Table 1); this was an unexpected finding in the case-control study set within the main unpredicted result and was inexplicable in terms of the study design. Rather, the main point about the case-control study results is that the intervention group itself did not improve, and the main problem is the pattern of results in which this negative finding sits alongside the positive expected finding of the community evaluation study. In this context, we consider three questions:What explains the different patterns of results in the two evaluation studies?Which was the appropriate evaluation strategy to test the effectiveness of the PACT project?Are there general implications for evaluating community-led health projects?

### 4.2. Differences in Study Populations and Sampling Strategies May Explain the Differing Results between the Two Studies

As noted, we found statistically significant differences in baseline scores between the two studies: the community-engaged sample had a higher baseline PHQ-9 scores, indicating more distress, compared with the intervention arm of the case-control study. This may explain why the former decreased while the latter did not, because of, for example, the measurement issue that lower scores have less scope for reduction. More broadly, the differences in baseline scores, with increased distress in the community sample compared with the case-control sample, suggests that these samples are drawn from differing populations in terms of the outcomes being assessed. The PHQ-9 and GAD-7 are used both extensively in clinical settings, but population norms are less researched. There are available population norms from large nationally representative German samples: Kocalevent and colleagues [35] found a PHQ-9 mean of 3.1 (s.d. 3.5) for women, and Löwe and colleagues [36] found a GAD-7 mean of 3.2 (s.d. 3.5) for women. The comparison of these estimates with the PACT samples baseline means (Table 1) suggests that the case-control sample was relatively closer to population means, as would be expected from recruitment from maternity clinics, while scores in the community-engaged sample were relatively high compared with population means.

These baseline differences in mental health markers are perhaps unsurprising given the objectives of the PACT project, namely to address local health inequalities by engaging women from local populations that typically have reduced access to statutory health services. As detailed elsewhere [21], PACT was successful in this respect; the project drew in mothers from local ‘hidden populations’, a term taken from research by local government referring to groups underrepresented in the census with low engagement with local services, poorer English language skills, a lack of awareness of services, shifting households, and immigration concerns. In the community study evaluation (n = 61), approximately 25% identified themselves as being from Nigeria and approximately 10% as Latin American. This contrasts with the 2011 census records which report only 4.7% Nigerians and 2.7% Latin Americans living in the borough of Southwark [37]. In fact, the 10% Latin American participation in PACT is an underestimate; there were, in fact, so many Latin American mothers wanting to join in, at varying stages of acquiring English, that a Spanish-speaking PACT sub-project was set up by the mothers and a Latin American parent champion. The community sample also comprised 25% self-identifying as Black African from Eritrea, 10% White British, and 10% White Other, mainly from continental Europe. Most mothers in the community study sample did not have English as their first language, and most were unemployed. By comparison, in the case-control study (n = 135), the self-declared ethnicity of the sample was 13% Black African, 25% White British, 20% White Other, and 14% Latin American. A majority of mothers had English as their first language, and only a fifth were unemployed.

Other demographic data suggest baseline differences in the samples in the two evaluation studies. For example, in employment status, 73.3% (N = 131) of women in the case-control study were in part-time or full-time employment, and 22.2% were unemployed. By contrast, for women in the community study evaluation, 34.4% (N = 61) were in part-time or full-time employment, and 62.3% were unemployed. Notwithstanding the lower levels of employment in the community study sample, this was not related to education. For data obtained on the highest educational level, 25.7% (N = 105) of women in the case-control study had university-level qualifications, and 41.0% (n = 61) in the community study evaluation sample had such qualifications.

The socio-demographic contrasts described above are consistent with the experiences of researchers, health professionals, and community members involved in PACT. Mothers engaged in the project via community organizing routes (several hundred, from which the evaluation sample of 61 was recruited) typically had different characteristics from the mothers engaged following a research protocol into the case-control evaluation. The former group of mothers was ethnically more diverse, had stronger linkages to existing institutions, such as faith groups and primary schools, and tended to be economically poorer, with many in temporary accommodation and/or with uncertain immigrant status and uncertain or no access to welfare support. 

The two evaluation studies also differed in terms of sampling frames and engagement methods. The sampling frame for the community study evaluation was mothers participating in PACT engaged by community organizing methodology, involving relationship building and emphasis on the benefits of participation, as described at the beginning of Section 2.2. This type of engagement led to participants being very motivated to access what the project had to offer and being inter-personally engaged. By contrast, the sampling frame for the case-control study was women attending community maternity clinics, recruitment was by research midwives following strict NHS research ethics guidelines, and the primary ‘ask’ was to help with research. There was no seeking out of other members of the community, such as isolated neighbors, who might benefit from participation in the project. Before informed consent to participate in the research, the women had no experience of PACT and there was no development of interpersonal relationships.

In summary, the community study participants and the case-control participants differed at baseline in important aspects and were engaged in completely different ways. Specifically, the community-engaged mothers had higher baseline levels of distress and were more socioeconomically disadvantaged. Following epidemiological literature on social determinants of health and health inequalities [3], we would expect a negative correlation between stress and wealth; socio-economically disadvantaged populations are relatively less resourced, tend to access statutory services less, have more stresses, and accordingly tend to have more distress. These important differences in mental health and health-related characteristics in the samples for the two evaluation studies of the PACT project may explain the difference in the patterns of results, one suggesting the benefits of participation, and the other not.

### 4.3. The Community Study Design Was the More Ecologically Valid Evaluation Strategy for the PACT Project

Given that the two PACT evaluation studies led to different results, one indicating benefits for participants, the other not, which one should we believe? A better question is this: which design is the most appropriate for evaluating the PACT project? From a research point of view, bearing in mind the higher strength of evidence that can accrue from controlled designs compared with non-controlled, the case-control study would be favored. That said, the application of the case-control design in the PACT evaluation in fact did not achieve its usual primary purpose, namely distinguishing whether or not a predicted positive uncontrolled effect size survived as a positive controlled effect size; rather, there was no positive uncontrolled effect size. In this sense, the case-control study delivered no information except the negative finding that participation in the project showed no benefits for the study sample. However, and this is the point we emphasize here, the controlled study lost the distinctive community-led engagement processes that were integral to the project and had to instead use researcher-led engagement processes, resulting in different sampling frames and samples. It was the community study evaluation strategy that retained community-organized engagement, integral to the PACT project, used to optimize engagement, especially of the relatively socioeconomically disadvantaged groups, aiming to reduce local health inequalities, and, in this sense, it is the more ecologically valid evaluation of the project.

### 4.4. Wider Implications for Evaluating Community-Led Health Projects

#### 4.4.1. Theorizing Community-Led vs. Researcher-Led Engagement

An ongoing debate exists as to whether the standard hierarchy of evaluation, with controlled designs towards the top, is suitable for complex interventions in community health. A first challenge is the requirement of the standardization of the intervention. A second challenge, the focus of this paper, relates to the standardization of the sampling methodology, especially for projects that are community-led. We have suggested, illustrated by the PACT case study, that there is a tension between the requirement of the standardization of the sampling methodology, in effect requiring that it be researcher led, and the recognition that, in community-led projects, the sampling pool and engagement methodology is definitely or likely to be out of researcher control. We reviewed in the Introduction a proposal by Hawe and colleagues [16] for overcoming the standardization of the intervention problem: a general model of intervention can be standardized, allowing adaptation to variable local conditions. We suggest that this approach can also be used to solve some of the second kind of standardization problems, the ones to do with sampling methodology. Methods of community-led engagement, while flexible and adaptable to local conditions, can be described in broad terms, as can local variations. The community engagement in PACT followed established community organizing methods, forging alliances between local civic and statutory organizations, establishing institutional and interpersonal relations towards a common purpose. Importantly, this abstracted description of local organizations, shared interests, goals, and implementation is generalizable from place to place. For example, in different localities, or for different populations, different organizations and different methods of providing social support, or health education content, might come into play. In this sense, it is possible to replicate the PACT project in other localities, and indeed this is currently being done.

However, while the generalizability problem can be resolved in this way, there is a further problem about community-led engagement and sampling methodology that is not solved, one which challenges more directly a fundamental assumption of the standard hierarchy of evaluation, namely the feasibility of constructing a control group. The problem, in brief, is that community-organized, community-led engagement relies purposefully on active social relationships and shared interests and cannot result then in nothing to do with either. Put more strongly, community-organized and community-led projects are more akin to *social campaigns* than scientific sampling. Indeed, Citizens UK often refers to its work in terms of campaigns, such as, for example, its long-running Living Wage campaign [38]. The community organized engagement methods explicitly rely on enthusiasm, encouragement, friendliness, group solidarity, and appeal to the interests and motivation of potential participants. Specifically, as we were aware when setting up the project, the distinctive characteristics of the social campaign-like community organizing approach to community engagement precluded the construction of a control group, since one cannot lead someone towards a project with persuasion and anticipation and then exclude them once they are assigned to the control group.

The characteristics of community-organized engagement processes methods, as outlined above, are quite different compared with the clinical trial methodology within which the hierarchy of evidence has been developed and refined. Clinical trials are typically set in hospitals with patients, the sampling frame is researcher-defined by a health marker (typically a diagnosis), and patients typically want treatment for their condition. This background of clinical trial methodology cannot be assumed in community health interventions, especially not in those that are community-led rather than researcher-led. The potential participants are typically not identified by a health marker, and there are not likely to be mutually agreed prior concepts around ‘treatment’ and ‘intervention’. On this point, the rejection of the term “intervention” by community stakeholders was evident in PACT’s co-production phase, as noted above (Section 2.2). Furthermore, if the community project aims to engage people who tend not to access statutory services, to be marginalized and isolated, as they often do, then engagement has to involve active seeking out and persuasion. This again is quite unlike the typical situation of a clinical trial with identified and self-identified patients, where involvement in the hospital/treatment system is already in place, and active seeking out and persuasion is scientifically inappropriate and probably prohibited by the ethics committee. Clinical trial methodology and its background assumptions make the construction of a control group feasible. Information leaflets have statements along the lines: ‘we want to know the benefits of this new treatment, whether there are any; if there are benefits, the group not receiving it in the trial will be offered it later, and anyway you will be receiving the best treatment as usual in the meantime.’ The whole idea, monitored by clinical research ethics committees, is to make sure that, to the best of current knowledge and belief, patients can safely opt into a trial not knowing which arm they will be assigned to, treatment or control, because they and their health condition are being well looked after either way. There is nothing analogous to all this in community settings where potential participants in a project or its evaluation are not identified (and self-identified) by a health marker. In its absence, there is no straightforward sense in the idea of community-led engagement into a health-related project, including not only engagement into the project but also into a control group.

To summarize, clinical trial methodology and the associated hierarchy of evidence valuing control conditions is itself a complex intervention, applied in complex systems, which can vary widely in factors that are relevant to its working assumptions and implementation. While it fits well with the clinical settings in which and for which it has been developed, it is problematic in community health projects, especially those that are community-led and aiming to include the often excluded.

Insofar as community organized engagement methods are, as we have suggested, incompatible with what is in the health sector a fundamental expectation of controlled evaluation, we believe it would be helpful to theorize these methods from a health sector perspective. We specifically consider health and clinical psychological perspectives on the PACT project’s primary health outcomes, anxiety and depression. Cognitive models of anxiety and depression implicate negative self-appraisals of coping capacity, self, and the future [39,40], which have some connections with the mechanisms of learned helplessness [41], and which are accordingly targets for intervention. We suggest that community-organized and community-led psychosocial engagement processes can be understood and theorized in such terms. They refer, of course, to individual-centred, psychological models of distress and need to be adapted in the present context for social circumstances. The health inequalities studies implicate a lack of economic resources and other processes of social exclusion in depressing perceived coping and raising levels of stress and stress-related conditions. Community-organized and led projects can target these mechanisms by increasing hope, social solidarity, support, and power. A small-scale plausible example of this in the development of the PACT project was in the pilot study, where the women came to support themselves (rather than asking for support from others) and, once in a cohesive group, felt able to actively seek out health education workshops from professionals—something they had not been able to do, or had not done, alone. 

#### 4.4.2. Optimizing Evaluation of Community Led Health Projects

We have suggested that community led health projects create severe challenges for the construction of a control group and evaluation using experimental or quasi-experimental designs. What, therefore, are the implications for evaluation of such projects? The rationale for evaluation, outlined in the Introduction, includes having confidence that the project does bring the desired benefits to participants and that it represents a good use of money intended for health improvement purposes—confidence that is required by communities themselves and by the health sector. Since the health sector is used to the standard hierarchy of evidence developed for clinical trial methodology, with experimental designs at the top, followed by quasi-experimental, with the rest being uncontrolled and in this respect sub-optimal, it is natural enough that they would expect or hope for this same epistemic framework to be applied also to community-led health interventions. In practice, however, very few community-led projects ever get to be evaluated using control groups, so there is something of an impasse here.

On the other hand, the health sector is used to funding new services that are evaluated, if at all, without controls. A very-large-scale example is the UK NHS Increasing Access to Psychological Therapies (IAPT) programme [42]. The IAPT services routinely collect pre-/post-intervention data, which has shown good results over the years, replicated in many services across the country. This ongoing practice evaluation supports confidence in the effectiveness of the IAPT services, but this confidence also derives from the original data that led to setting up the new services, namely that there was an unmet population need for treatment for common mental problems, anxiety and depression, and—the crucial point—the fact that there was an accumulating evidence base from clinical trials, with strength of evidence high up the hierarchy, that model-based psychological therapies were effective for these conditions. In short, simple pre-/post-intervention data suggesting improvement may be good enough in the roll-out of a service, provided the intervention has been tested successfully under controlled conditions. Applied to community health projects, the inference would be that pre-/post-participation outcome data should always be collected, on a sample of participants (with sampling of course independent of response), requiring a clear statement of intended health-related outcomes and the use of standardized measures to assess them, as well as the requirement that the components of the project—specific interventions—should have a reasonably strong theoretical and evidential basis in relation to those outcomes. We supposed that these conditions of reasonably strong theoretical and evidential basis were satisfied by the PACT core component interventions of social support and health education, in relation to distress-related conditions and health literacy, as well as applying to the method of engagement itself. It is unlikely that all these conditions of good enough evaluation of a health-related community project can be put in place without the expertise of health professionals and health researchers, just as, conversely, a health-related project is unlikely to engage communities, especially those that tend to access statutory services less, without the expertise of professional community workers, such as community organizers.

## 5. Conclusions

While controlled designs for evaluating the effectiveness of health interventions may be preferred by research funders, policymakers, and commissioners, consistent with the usual hierarchy of strength of evidence, our attempt to implement this in a community-led health project in South London encountered severe challenges. We found that a standardized case-control design study with a researcher-defined sampling pool and researcher-led engagement resulted in very different samples and outcomes compared with the more ecologically valid community-led engagement process evaluated in a simple pre-post design. While there are severe challenges to evaluating a community-led health intervention using a controlled design, the measurement of pre-/post-participation changes in well-defined health outcomes should typically be a minimum evaluation requirement, and confidence in attributing causation of any positive changes to participation can be increased by the use of interventions in the project and in the engagement process itself that have a credible theoretical and empirical basis.

## Figures and Tables

**Table 1 ijerph-17-02523-t001:** About here.

Study & Sample (Number)	PHQ-9		*p*-Value	GAD-7		*p*-Value
	Baseline Mean (SD)	Follow-Up Mean (SD)		Baseline Mean (SD)	Follow-Up Mean (SD)	
Community Study sample (6 months to follow-up)						
Whole sample (baseline N = 61; follow-up n = 58)	7.66 (6.37)	4.83 (4.15)	*p* < 0.000 (t = 3.78, df = 57)	6.87 (5.62)	4.76 (3.85)	*p* = 0.001 (t = 3.36, df = 57)
	Case-control study {10.5 months to follow-up}					
Intervention arm (n = 68)	4.50 (3.93)	4.86 (4.47)	ns	4.22 (3.52)	4.50 (4.07)	ns
Control arm (n = 67)	5.31 (4.64)	4.10 (4.08)	*p* = 0.006 (t = 2.86, df = 66)	4.90 (4.66)	4.57 (4.25)	ns

Table 1: Descriptive statistics for the PHQ-9 and GAD-7 for the community study and the case-control study (intervention arm and control arm): numbers, group means (standard deviations) at baseline and follow-up, and paired *t*-test significance level.

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
