# Peer review of "Evaluation of a Community-Led Intervention in South London: How Much Standardization Is Possible?"

_ijerph, 2020, doi:10.3390/ijerph17072523_

Round 1

Reviewer 1 Report

I appreciate the difficulty in writing this paper.  The group did a nice job describing the study and challenges.  For interventions that are meant for the communities rather than clinics, I tend to advocate for community-led interventions rather than researcher-led intervention because clinical trials can not seem to account for heterogeneity and dynamics of community relationships. So we might as well study the interventions in the environment that they are supposed to be used and learn as much as we can from that community.  It saves time and money.  I really enjoy reading this manuscript.   Nice job.

Reviewer 2 Report

I understand the approach that the authors wanted to give to the work however I remain of the idea that the small changes required are necessary. I believe that while the quantitative cutting work remains, without having to open a debate, the authors should make some considerations both in the introduction part in relation to the stakeholders and in the discussion highlighting the limits of the work
I therefore repropose what has already been highlighted above as suggestions for improving work.

In the introductory part to line 55 I think it is important to mention some recent works which, by proposing the photovoice technique as a working method, manage to involve and synergize stakeholders with members of the community. In this regard, a recent work
“Sustainability 2019, 11(23), 6822; https://doi.org/10.3390/su11236822
“Article The Collapse of the Morandi Bridge in Genoa on 14 August 2018: A Collective Traumatic Event and Its Emotional Impact Linked to the Place and Loss of a Symbol by Nadia Rania *,Ilaria Coppola, Francesco Martorana and Laura Migliorini

“To evaluate their research project, the authors completely neglect the qualitative approach. I think a mention of this during the discussion could be important for research developments. Rather than building two groups that have such a different base, perhaps integrating a qualitative approach with an interview and / or focus group to their measures could help to better understand and interpret the data found.”

Author Response

This manuscript is a resubmission of an earlier submission. The following is a list of the peer review reports and author responses from that submission.

Round 1

Reviewer 1 Report

The purpose of article is to highlight challenges in evaluating a community-led intervention to improve health outcomes of new mothers and their children through social support and health education.  More specifically, the authors try to explain reasons for the observed differences in the results of the two different study designs to evaluate this intervention: a community-led vs. researchers-led sampling frames.  In the community-led study, participants showed statistically significant benefits but in the researchers-led ‘case-control’ design, no significant results were found.  The authors concluded that the different sampling/recruiting strategies which led to differences in study populations may explain the different results between the two studies and further discussed general implications for evaluation community-led health projects.

Lessons learned from this project could be useful to other researchers engaged in community-led health projects.  The article is well-written and I only have a few minor comments.

I’m not sure what the word limit for this journal but the article seems long. While the introduction/background was interesting, the lengthy intro was a distraction to get to the objectives of the manuscript. In the discussion of the study #1, the authors mentioned that participants were recruited through the maternity clinics but the recruitment process for study #2 on page 8 was less clear. Then we finally found that out on page 11.  It appears the authors were trying to save that information for the discussion.  I think  it is fine to talk about the different sampling frame upfront and discussed it again in the discussion. I appreciate the acknowledgement of the cluster randomization design and why they couldn’t do it because of limited funding. The entire manuscript could benefit from more trimming of the text in all sections (except the results could use more data). While the writing itself is good, there appears to be quite a bit of repetitions.  For each of the important point in the discussion for instance, it can probably be written more succinctly.  The results section could use more explanation and data from negative findings.  Negative findings are just as important as positive findings so there’s no need to include the results in the appendix.

Reviewer 2 Report

Thank you for opportunity for reviewing this paper “Evaluation of a community-led intervention in South 2 London: how much standardization is possible?”  The authors did good job in comparing pros and cons of using a controlled design in a community. I have some thought to add more for your consideration. One appropriate research design for community setting is action research. As we know, action research can be categorized into three types, depended on who is the researcher?: 1) (Professional) action research: researcher is the academic officer; 2) (People) participatory action research: researchers are people in a community and academic officer; 3) People action research: researcher is people in a community. My thought is that which research type or research design should be used in the community will depend on the strength of that community. In addition, the purpose of conducting research in a community is usually for solving the problem of that community. Therefore, the limitation of generalization will occur.

Reviewer 3 Report

This is an intriguing, highly relevant report concerning implementation of community engaged research. The take home point that results differ, based on the recruitment process, is important to document. The premise fits with what many have encountered and know from experience with community engagement and research- mainly there is high value in peer network recruitment, interaction and engagement. This concept should be well documented and shared in the literature to advance effectiveness of conducting community engaged research and implementation science.

The background and research design are well described and rationale and argument effectively presented. I would like to see this discussion in the literature to consider in design of studies. However, some serious concerns should be addressed.

Basic design of study: The rational of the two studies reported is made clear. Difference in changes in PDQ-9 and GAD-7 measures of maternal mental health differ for researcher-enrolled study participants than for the community-enrolled participants is the main outcome data of the paper. While the community-engaged participants change significantly from pre- to post in mental health indicators, the researcher-enrolled group does not (Table 1). An explanation that must be discussed is what in the process of the researcher-enrollment accounts for features of the selected women who baseline begins at a lower level on the assessments used. This might be explained by several factors including the requirement that women enrolled for the study must engage with the academic researchers, complete the necessary forms and other requirements to be part of a "research study". This might attract certain participants and unintentionally exclude others.

No analyses are presented in the current manuscript of demographic data such as participant education level, income, etc. The community-engaged women were required to speak English well enough to complete forms. However, invitation by a peer or community leader may have resulted in participants with different initial levels of maternal mental health and preparation. That the baseline for the two groups differ substantially, suggests that some selection criteria, not yet identified, was at play. This is important.

One could argue that although all participants are women who are mothers, these groups differ from the start. For a useful analogy, an improved spelling competency test is given to two participant groups of similar age from the same neighborhood. One group was composed of participants who had completed 6th and 7th grade. The other group was composed of a range of participants whose highest grade completed ranged between 3rd grade and 7th grade. Although they received the same intervention, and ended up at the same level of spelling competency, data for the second group likely would differ from that of the first group if presented only as change in spelling competency because of the variation in starting point. Such was not the intent of the study, but some factor(s) in the selection or intervention process results in a baseline level that differs for the community engaged group. To support the premise and major result, one would need to start with similar levels in the feature measured (maternal mental health) to then ask if the two groups change from the similar level over time. If I read presented outcomes correctly, the opposite is reported in Table 1. A possible explanation is that the researcher-recruited group (likely due to something in the process) had some factor within the selection that makes them different as indicated by a lower measured baseline PDQ-9 and GAD-7 levels.

Are there demographic data on education? These, or other insights to account for the baseline difference, should be included. 

I appreciate the thorough introduction and background to establish the issue. This might make a highly useful review if additional data requested is not readily available.

The Additional Material in Appendix A must be edited well to reduce wordiness and correct minor errors.

Summary: The paper seeks to highlight an important factor in community-engaged intervention that should be considered in design of implementation research – mainly understanding the importance of academic researcher-led or community initiated and community-led studies on engagement, retention and uptake of intervention content. It brings the questions, what is required for effective progress in moving health and behavior intervention into community. What is the balance between scientific rigor and expanding the reach into and effectiveness with communities?

Reviewer 4 Report

The manuscript is very interesting to the reader and challenging for researchers. I provide below to the authors some suggestions for improving their work:

In the introductory part to line 55 I think it is important to mention
some recent works which, by proposing the photovoice technique
as a working method, manage to involve and synergize
stakeholders with members of the community.
In this regard, a recent work

Sustainability 201911(23), 6822; https://doi.org/10.3390/su11236822

Article The Collapse of the Morandi Bridge in Genoa on 14 August 2018: A Collective Traumatic Event and Its Emotional Impact Linked to the Place and Loss of a Symbol by Nadia Rania *,Ilaria Coppola,Francesco Martorana andLaura Migliorini   To evaluate their research project, the authors completely neglect the qualitative approach. I think a mention of this during the discussion could be important for research developments. Rather than building two groups that have such a different base, perhaps integrating a qualitative approach with an interview and / or focus group to their measures could help to better understand and interpret the data found.  
